# In Situ Doping of Nitrogen in <110>-Oriented Bulk 3C-SiC by Halide Laser Chemical Vapour Deposition

**DOI:** 10.3390/ma13020410

**Published:** 2020-01-15

**Authors:** Youfeng Lai, Lixue Xia, Qingfang Xu, Qizhong Li, Kai Liu, Meijun Yang, Song Zhang, Mingxu Han, Takashi Goto, Lianmeng Zhang, Rong Tu

**Affiliations:** 1State Key Laboratory of Advanced Technology for Materials Synthesis and Processing, Wuhan University of Technology, 122 Luoshi Road, Wuhan 430070, China; 19951114@whut.edu.cn (Y.L.); xlxwhutmail@163.com (L.X.); qizhongli@whut.edu.cn (Q.L.); liyangmeijun@163.com (M.Y.); superkobe0104@gmail.com (S.Z.); goto@imr.tohoku.ac.jp (T.G.); lmzhang@whut.edu.cn (L.Z.); 2School of Optical and Electronic Information, Huazhong University of Science and Technology, No. 1037 Luoyu Road, Wuhan 430070, China; 2019508016@hust.edu.cn; 3Materials Science and Engineering, Wuhan University of Technollogy, 122 Luoshi Road, Wuhan 430070, China; victor_liu@whut.edu.cn; 4R & D, Ibiden Co., Ltd., 1-1 Kitagata, Ibigawa-cho, Ibi-gun, Gifu 501-0695, Japan; han_mingxu@ibiden.com; 5New Industry Creation Hatchery Center, Tohoku University, Sendai 980-8579, Japan

**Keywords:** N-doped &lt, 110&gt, -oriented 3C-SiC bulk, preferred orientation, conductive SiC, halide laser CVD

## Abstract

Doping of nitrogen is a promising approach to improve the electrical conductivity of 3C-SiC and allow its application in various fields. N-doped, <110>-oriented 3C-SiC bulks with different doping concentrations were prepared via halide laser chemical vapour deposition (HLCVD) using tetrachlorosilane (SiCl_4_) and methane (CH_4_) as precursors, along with nitrogen (N_2_) as a dopant. We investigated the effect of the volume fraction of nitrogen (*ϕ*_N2_) on the preferred orientation, microstructure, electrical conductivity (*σ*), deposition rate (*R*_dep_), and optical transmittance. The preference of 3C-SiC for the <110> orientation increased with increasing *ϕ*_N2_. The *σ* value of the N-doped 3C-SiC bulk substrates first increased and then decreased with increasing *ϕ*_N2_, reaching a maximum value of 7.4 × 10^2^ S/m at *ϕ*_N2_ = 20%. *R*_dep_ showed its highest value (3000 μm/h) for the undoped sample and decreased with increasing *ϕ*_N2_, reaching 1437 μm/h at *ϕ*_N2_ = 30%. The transmittance of the N-doped 3C-SiC bulks decreased with *ϕ*_N2_ and showed a declining trend at wavelengths longer than 1000 nm. Compared with the previously prepared <111>-oriented N-doped 3C-SiC, the high-speed preparation of <110>-oriented N-doped 3C-SiC bulks further broadens its application field.

## 1. Introduction

Silicon carbide (SiC) is a promising electronic material, which has been applied in various fields due to its excellent mechanical properties, chemical stability, and high durability [1,2]. Although SiC is an intrinsic semiconductor, a high electrical conductivity (*σ*) is a necessary requirement for certain applications, such as solar cells [3], electromagnetic shielding materials [4], pressure sensors [5], and electro-discharge machining [6]. In situ doping via chemical vapour deposition (CVD) is a common method to improve the electrical conductivity of intrinsic semiconductors and allow its application in various fields, such as diamond, ZnO, TiO_2_, etc. [7,8,9]. In recent decades, a great deal of work has been focused on the preparation of N-doped SiC by adding NH_3_ [5,10] or N_2_ [11,12] during CVD, achieving *σ* values from 0.1 to 10^3^ S/m. However, the deposition rates (*R*_dep_) were limited to values between 0.1 and 10 μm/h [10,11,12,13,14]. Due to the low deposition rate, conventional CVD is usually effective for preparing SiC films but not bulk crystals, which greatly limits the applications of doped SiC.

Our research group has recently developed an efficient halide laser CVD (HLCVD) process to fabricate dense and pure 3C-SiC bulks with high *R*_dep_, including transparent 3C-SiC bulks with low defect density and highly preferred orientations [15,16,17]. The maximum *R*_dep_ of 3C-SiC prepared by HLCVD reached 3600 μm/h, which is 10 to 10^3^ times greater than that of conventional CVD. Preferred orientation has a great influence on the properties of corresponding crystals, for example, <111>-oriented 3C-SiC has higher hardness than that of <110>- and <111> co-oriented [15], <110>- 3C-SiC can be used in biosensors by modifying it using an organic molecule [18]. Although 3C-SiC prepared by CVD usually exhibit <111> and <110> preferred orientations, the doped 3C-SiC obtained in previous studies showed an almost exclusively <111> orientation. Nevertheless, the <110>-oriented 3C-SiC may be a more sensitive structural material and could be applied to pressure sensors in microelectromechanical systems (MEMS) [19], due to its lower Young’s modulus (about 350 GPa) than that of the <111>-oriented 3C-SiC (about 500 GPa).

NH_3_ is an effective and widely used precursor in N-doped SiC, but in the chloride precursor system, especially when the flow rate of the precursor is large, NH_3_ reacts violently with chloride to form ultra-fine powdery by-products and easily block the pipeline and vacuum pump. In addition, large amounts of Si-N bonds are easily generated due to the high reactivity of NH_3_. On the contrary, although the reaction activity of nitrogen is low, N_2_ is often used as the precursor of semiconductor doping because of its good safety and large flow regulation range. In our previous study, we showed that laser CVD can rapidly produce highly <110>-oriented 3C-SiC at a relatively low temperature (approximately 1523 K) [15]. To obtain a <110>-oriented 3C-SiC bulk with high electrical conductivity at high *R*_dep_, in situ nitrogen doping was conducted by adding nitrogen during the preparation of 3C-SiC via HLCVD. In this study, we prepared thick, conductive, and N-doped <110>-oriented 3C-SiC bulks and investigated the effect of the volume fraction of N_2_ (*ϕ*_N2_) on their preferred orientation, deposition rate, electrical conductivity, and optical transmittance. This study provides a new promising route for the rapid fabrication of various doped bulk semiconductors with specific preferred orientation.

## 2. Experimental

A cold wall HLCVD apparatus was developed to fabricate N-doped 3C-SiC bulks. Figure 1 shows the diagram of the HLCVD apparatus. Graphite discs (IGS-743, *ϕ =* 15 mm × 1 mm, Sankyo Carbon, Tokyo, Japan) were used as substrates and heated on a heating stage at a temperature of 773 K before deposition. A diode laser beam (InGaAlAs, *λ* = 1060 nm, BEIJING ZK Laser Co., Ltd., Beijing, China) was introduced into the chamber through a quartz window and its diameter was enlarged to 20 mm by a lens to cover the entire substrate. The deposition temperature (*T*_dep_) was measured with a pyrometer (2MH-CF4, Optris GmbH, Berlin, Germany) and automatically controlled at 1623 K by a computer. The error in the surface temperature of the entire graphite substrate was ±5 K. SiCl_4_ and CH_4_ were used as precursors and N_2_ as dopant. Liquid SiCl_4_ was evaporated into a gas by an evaporator at a temperature of 353 K before being carried into the CVD chamber. The flow rates of SiCl_4_, CH_4_, and H_2_ were fixed at 600, 200, and 1200 sccm, respectively (the flow rate of SiCl_4_ was converted from liquid to gas units), whereas the flow rate of N_2_ was controlled to 5% to 30% of the total gas. The total pressure (*P*_tot_) and deposition time were set to 4 kPa and 15 min, respectively. The distance between the graphite substrate and the nozzle was 30 mm. The nozzle temperature was set to 473 K. An exhaust gas treatment system composed of a cold trap filled with liquid nitrogen, an activated carbon filter, and an NaOH spray scrubber was used for the disposal of the harmful acidic intermediates. Table 1 summarises the main deposition parameters used for the preparation of N-doped 3C-SiC bulks.

The crystalline phases of the deposited samples were analysed by *θ*-2*θ* X-ray diffraction (XRD) with Cu K_α_ radiation at 40 kV and 40 mA (Ultima III, Rigaku, Tokyo, Japan). The microstructure and thickness of the deposits were inspected by field emission scanning electron microscopy (FESEM, Quanta-250, FEI, 20 kV, Houston, TX, USA). Raman spectra were measured by a LabRam HR800 Ev (Horiba, Tokyo, Japan) spectrometer equipped with a 532 nm He-Ne laser. The transmittance was measured in the wavelength range of 250 to 2000 nm using a UV-Vis-near infrared (NIR) spectrophotometer (Lambda 750 S, PerkinElmer, Waltham, MA, USA). Before testing the optical transmittance of 3C-SiC, the thick deposits were removed from the graphite substrate by a diamond grinding disc and polished into a mirror surface on both sides using diamond slurries of diameters 9, 3, and 1 μm. The final thicknesses of the 3C-SiC bulks were approximately 250 μm. The electrical conductivity and Hall coefficient of the 3C-SiC bulks were measured by a self-assembled electric and magnetic performance testing system. The elemental composition of the N-doped 3C-SiC bulks was analysed by X-ray photoelectron spectroscopy (XPS, ESCALAB 250Xi, Thermo Fisher Scientific, MA, USA). To prevent surface contamination from affecting the results, all samples were argon-etched prior to testing. The Seebeck coefficients were determined using a portable tester (PTM-3, Jouleyacht, Wuhan, China).

## 3. Results and Discussion

Figure 2a shows typical XRD patterns of the 3C-SiC bulks prepared by HLCVD with *ϕ*_N2_ values up to 30%. No traces of second phases were detected in the XRD patterns. A small shoulder observed at 2*θ* ~ 33.8° could be attributed to planar defects [20,21,22]. All specimens deposited at *T*_dep_ = 1623 K and *P*_tot_ = 4 kPa exhibited a <110> preferred orientation. Figure 2b shows the Lotgering factors (*F*_hkl_) and lattice constants (*a*) of the N-doped 3C-SiC samples. *F*_hkl_ was calculated from Equation (1) [23]:(1)$$F_{hkl}=\left(P_{hkl}-P_{0}\right)/\left(1-P_{0}\right)$$
where *P*_hkl_ and *P*_0_ are the ratios of the peak intensity of the (*hkl*) planes to the sum of the intensities of all peaks for the films (*P*_hkl_) and powders (*P*_0_), respectively. The values of *F*_hkl_ range from negative (orientation along the other axes) to 0 (random) to 1 (perfect orientation). The *F*_110_ value of undoped 3C-SiC was 0.774 and increased with increasing *ϕ*_N2_, reaching 0.945 at *ϕ*_N2_ = 30%. To the best of our knowledge, this is the first report of N-doped 3C-SiC with <110> orientation. According to Refs. [24,25], <111>-oriented 3C-SiC shows a polar crystal structure containing Si {111} and C {-1-1-1} planes. The substituted N atoms mostly occupy the C sites in the SiC lattice. However, the adsorption of C or N on Si atoms along the Si-terminated surface of <111>-oriented 3C-SiC grains during the growth is relatively weak. With increasing *ϕ*_N2_, the site competition between C and N atoms becomes more intense, which may reduce the growth rate of the <111> orientation and result in the increase of *F*_110_. The lattice constant (*a*) of 3C-SiC decreased from 4.3707 Å for the undoped sample to 4.3640 Å at *ϕ*_N2_ = 30%. Nitrogen doping usually results in the shrinkage of the SiC lattice, due to the substitution of N atoms at the C sites and the smaller covalent radius of N (0.70 Å) than that of C (0.77 Å) [13].

Figure 3 displays the surface (a–e) and cross-sectional (f–j) morphologies of 3C-SiC bulks prepared with various *ϕ*_N2_ values. The undoped 3C-SiC bulk sample shows a cauliflower-like surface structure (Figure 3a) and a columnar cross section (Figure 3f). The cross-sectional morphology of the N-doped 3C-SiC samples are still columnar (Figure 3g–j), while the surface morphology (Figure 3b–e) changed with increasing *ϕ*_N2_. The aggregate particle size decreased from approximately 250 μm (undoped sample) to 60 μm (*ϕ*_N2_ = 30%), and the morphology gradually changed from cauliflower to fourfold-symmetric pyramid microstructure. The pyramidal grains were considered to be the final stage of the twin-plane propagation model previously proposed for fully <110>-oriented 3C-SiC [26]. The change of morphology indicates the enhancement of <110>-orientation. However, it is not a complete pyramid structure in Figure 3e, indicating that 3C-SiC is still not completely <110>-oriented when nitrogen volume fraction reaches 30%, which is consistent with the XRD results. Figure 4 shows the dependence of the deposition rate on *ϕ*_N2_. The *R*_dep_ values were calculated from the deposition time and film thickness. With increasing *ϕ*_N2_, *R*_dep_ decreased from 3000 μm/h for the undoped sample to 1437 μm/h at *ϕ*_N2_ = 30%. As the substitution of N atoms mostly occurs at the C sites in the SiC lattice, the dopant may occupy the surface sites of the carbon-containing group, leading to a decrease in the deposition rate [27].

Figure 5 shows the Raman scattering spectra of N-doped 3C-SiC bulks deposited at *T*_dep_ = 1623 K with different *ϕ*_N2_ values. The characteristic Raman modes of pure 3C-SiC primarily consisted of transverse optical (TO) and longitudinal optical (LO) phonons at 796 and 972 cm^−1^, respectively [28]. This is consistent with the Raman characteristics of the undoped 3C-SiC bulk. However, for *ϕ*_N2_ > 5%, the shape of the LO peak of the 3C-SiC bulks showed a marked change, due to Raman scattering from LO phonon-plasmon coupled (LOPC) modes. In a polar semiconductor, the free carrier plasma interacts with the LO lattice vibrations (phonons) via macroscopic electric fields. As a result of this interaction, instead of a pure plasmon and a pure LO phonon, a coupled plasmon-phonon mode appears in the spectrum [29,30]. Yugami et al. studied the appearance of the TO and LO peaks in N-doped 3C-SiC, and suggested that the increase in carrier concentration due to the incorporation of N was closely related to the LO peak. The LO peak shifted toward high frequencies and became broader with the decrease in peak intensity as the carrier concentration increased. In contrast, the frequency and full width at half maximum (FWHM) of the TO peaks did not change with the carrier concentration [31]. In this study, the frequency of the LO peak gradually shifted from 972 cm^−1^ to higher frequencies with increasing *ϕ*_N2_. In particular, the frequency of the LO peak reached 997 cm^−1^ for *ϕ*_N2_ = 30%.

Figure 6 displays the transmittance in the UV-Vis-NIR range of N-doped 3C-SiC bulks prepared at *T*_dep_ = 1623 K and *P*_tot_ = 4 kPa, with different *ϕ*_N2_ values. The photographs of the bulks, also shown in Figure 6, reveal that they had a translucent appearance. The differences in the transmittance characteristics of the undoped and N-doped 3C-SiC bulks can be divided into three regions [32,33]. In the low-wavelength region (below 500 nm), the fundamental absorption (FA) of N-doped 3C-SiC shifted to high wavelengths; in other words, the bandgap of these samples was shifted to lower energies (“bandgap shrinkage”). In the visible region, the slope of the transmittance curve of N-doped 3C-SiC was lower than that of undoped 3C-SiC, which caused the apparent colour of the doped SiC to change from yellow to green. The overall transmittance decreased with increasing *ϕ*_N2_, resulting in a darker apparent colour of the bulks. In addition, no absorption bands in the below-bandgap region (BGGA), which are usually present in the transmittance spectrum of N-doped α-SiC, were observed for N-doped 3C-SiC, because of the absence of nitrogen donors at inequivalent lattice sites in 3C-SiC [34]. In the NIR region, the transmittance of undoped 3C-SiC increased with increasing wavelength, whereas that of N-doped 3C-SiC decreased; this may be due to free carrier absorption (FCA) and is considered to be an intravalence band transition. The peaks between 850 and 950 nm are generated by the switch between the detector and the light source in the spectrophotometer [35].

Figure 7a shows the electrical conductivity (*σ*), carrier concentration (*n*), and electron mobility (*μ*) of the N-doped 3C-SiC bulks. The *σ* value of the undoped 3C-SiC sample was 7.1 Sm^−1^, and the corresponding intrinsic carrier concentration was 5.7 × 10^16^ cm^−3^, which is higher than that reported in the literature (~10^15^ cm^−3^) [13,31]. This means the undoped 3C-SiC may be an unintentionally N-doped sample. The *σ* value of the N-doped 3C-SiC bulks first increased and then decreased with increasing *ϕ*_N2_, reaching a maximum of 740.7 Sm^−1^ at *ϕ*_N2_ = 20%. When *ϕ*_N2_ was further increased, the conductivity showed a slight decrease. Simultaneously, the *μ* value of the 3C-SiC bulks decreased from 8.37 to 0.39 cm^2^/V⋅s with increasing *ϕ*_N2_. The decrease in mobility may be attributed to the decrease in crystallinity or the formation of amorphous Si_3_N_4_ in the 3C-SiC bulks at high *ϕ*_N2_ [10]. The corresponding *n* values increased with increasing *ϕ*_N2_, and the highest *n* of 3.3 × 10^19^ cm^−3^ was obtained for the N-doped 3C-SiC bulk deposited at a *ϕ*_N2_ = 30%. Figure 7b displays the elemental compositions of the 3C-SiC bulks. A stoichiometric 3C-SiC composition was obtained for the undoped sample, and the concentration of C atoms decreased slightly as *ϕ*_N2_ increased. In addition, the nitrogen concentration was two to three orders of magnitudes higher than the carrier concentration. This indicates that most nitrogen atoms are not activated and may be incorporated into the grain boundaries in the form of amorphous nitrogen complexes [13]. Figure 7c shows the Seebeck coefficient of the 3C-SiC bulks at room temperature. All 3C-SiC bulks investigated in this study showed a negative Seebeck coefficient, implying *n*-type conduction. In addition, the absolute value of the Seebeck coefficient decreased with increasing *ϕ*_N2_. The Seebeck coefficient and electrical conductivity usually follow an opposite trend as a function of the carrier concentration [36,37]. Figure 7d illustrates the temperature dependence of the electrical conductivity. The *σ* values of all 3C-SiC bulks increased with increasing temperature in the range of 295 to 775 K. The activation energy of *σ* was 0.08 eV for the undoped 3C-SiC and decreased to 0.03 eV at *ϕ*_N2_ = 30%, indicating the presence of a shallow donor level below the conduction band (with energy *E*_N_ = 0.03–0.08 eV) in the 3C-SiC bulks [38]. The slight decrease in activation energy with increasing *ϕ*_N2_ has been reported in several studies [5,13,39], and is generally attributed to the decrease in average potential energy of an electron [40].

Figure 8 compares the electrical conductivity and deposition rate of the N-doped 3C-SiC samples prepared in this study with those reported in the literature [5,10,11,12,13,14,41]. Since most previous studies only provide approximate deposition rates, we assume that they involve a constant deposition rate. Most doped 3C-SiC samples were prepared via conventional CVD using NH_3_ as dopant, and exhibited <111> but not <110> orientation. The electrical conductivities varied between 33 and 1 × 10^4^ Sm^−1^ while the deposition rates were in the 0.4–18 μmh^−1^ range. The N-doped 3C-SiC bulks prepared via HLCVD in this study exhibited predominant <110> orientation, high deposition rates (1430 to 1670 μmh^−1^, between 80 and 4000 times higher than those reported in the literature), and high electrical conductivities (7.4 × 10^2^ Sm^−1^).

## 4. Conclusions

Halide laser chemical vapour deposition is a promising method for obtaining 3C-SiC bulks with predominant <110> orientation and high electrical conductivity at high deposition rates. The degree of <110> orientation of 3C-SiC increased with increasing nitrogen content and became nearly complete at *ϕ*_N2_ = 30%, with a Lotgering factor (*F*_220_) of 0.945. Upon increasing *ϕ*_N2_ to 30%, the electrical conductivity and carrier concentration of the N-doped 3C-SiC bulks increased from 7.1 to 740 Sm^−1^ and from 5.7 × 10^16^ to 3.3 × 10^19^ cm^−3^, respectively. While the <110>-oriented 3C-SiC bulks were translucent, upon nitrogen doping their transmittance decreased with increasing *ϕ*_N2_ and also with increasing wavelength. Finally, the *R*_dep_ value decreased with increasing *ϕ*_N2_, reaching 1437 μmh^−^^1^ at *ϕ*_N2_ = 30%, which is 80 to 4276 times higher than that of reported in other studies.

## Figures and Tables

**Figure 1 materials-13-00410-f001:**
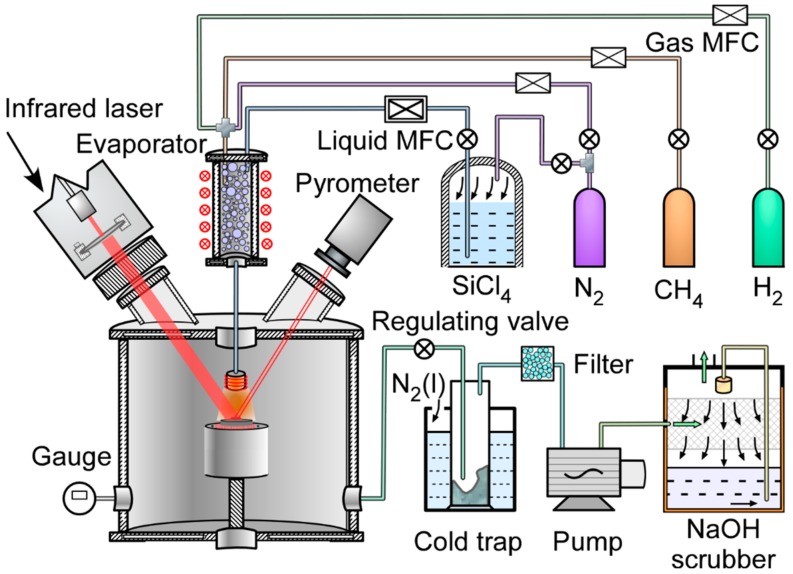
Schematic illustration of the HLCVD apparatus.

**Figure 2 materials-13-00410-f002:**
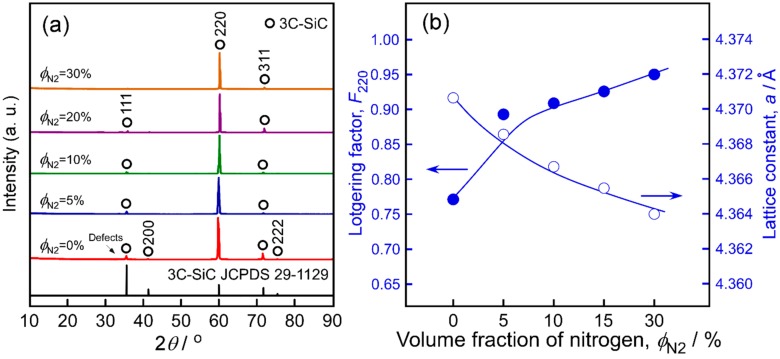
Crystallographic properties of N-doped 3C-SiC prepared with different *ϕ*_N2_ values. (**a**) XRD patterns; (**b**) Lotgering factors *F*_110_ and corresponding lattice constants.

**Figure 3 materials-13-00410-f003:**
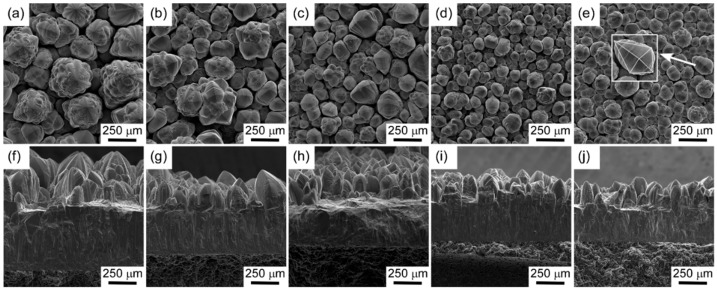
Surface and cross-sectional microstructure of N-doped 3C-SiC bulks obtained at *T*_dep_ = 1623 K and *P*_tot_ = 4 kPa, with *ϕ*_N2_ = 0% (**a**,**f**), 5% (**b**,**g**), 10% (**c**,**h**), 20% (**d**,**i**), and 30% (**e**,**j**).

**Figure 4 materials-13-00410-f004:**
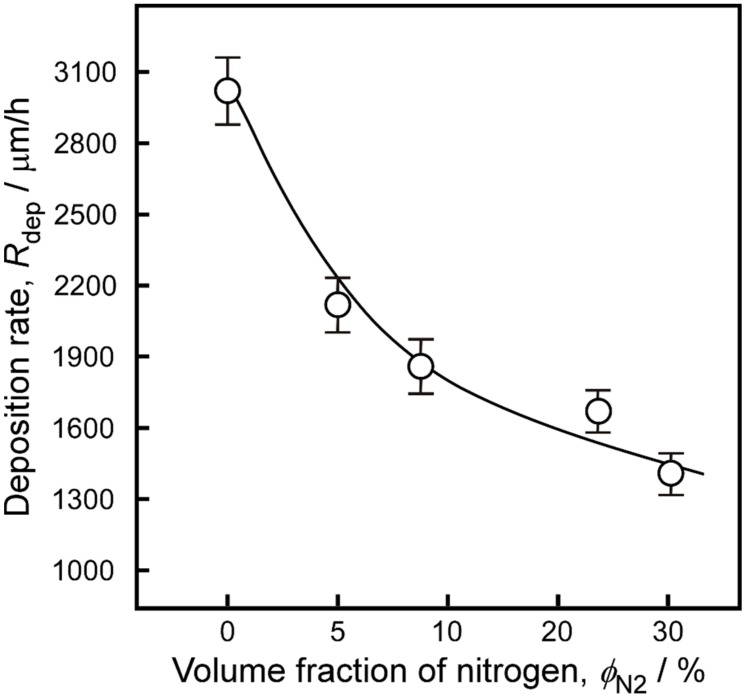
*R*_dep_ values of N-doped 3C-SiC bulks prepared with different *ϕ*_N2_ parameters.

**Figure 5 materials-13-00410-f005:**
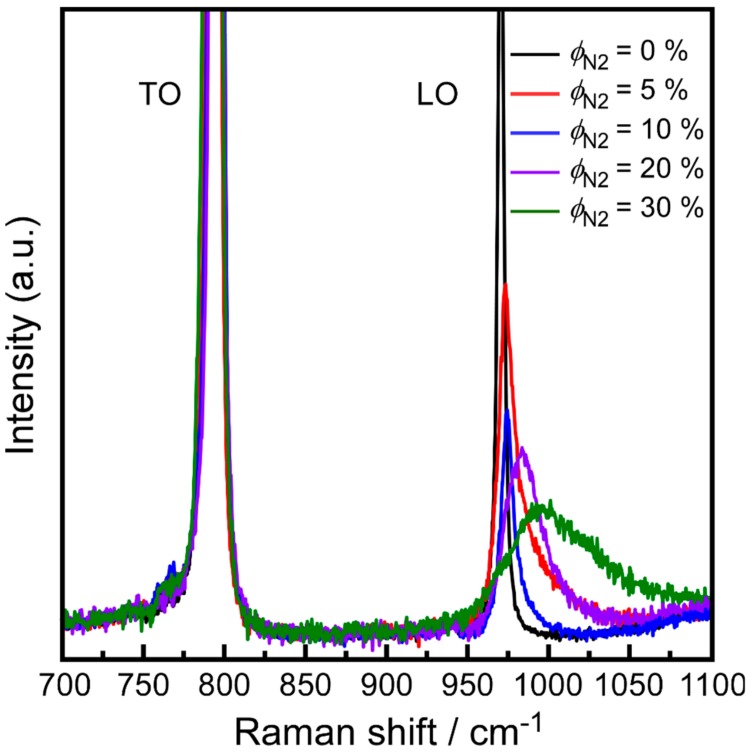
Raman spectra of N-doped 3C-SiC bulks grown with different *ϕ*_N2_ values.

**Figure 6 materials-13-00410-f006:**
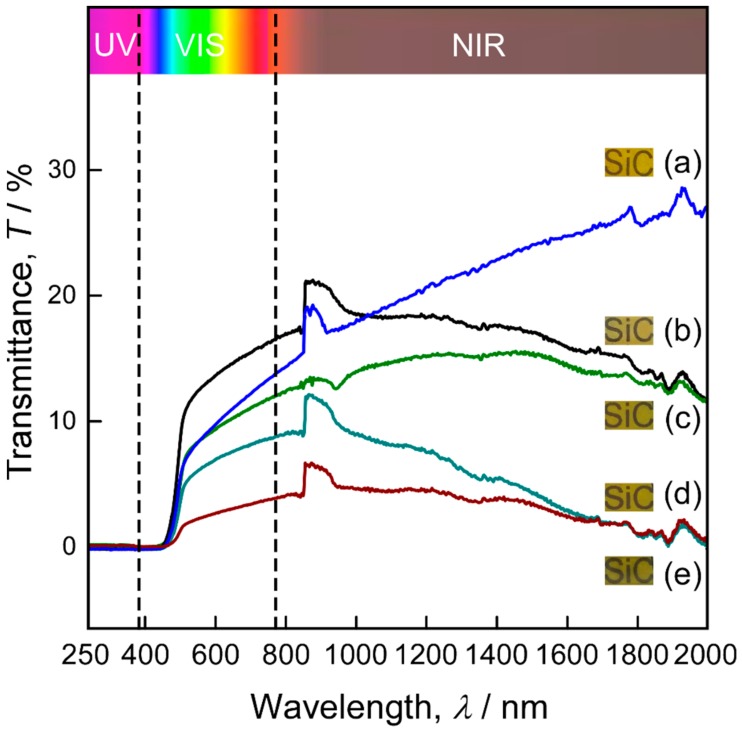
Photograph and optical transmission curves of mirror-polished N-doped 3C-SiC bulks.

**Figure 7 materials-13-00410-f007:**
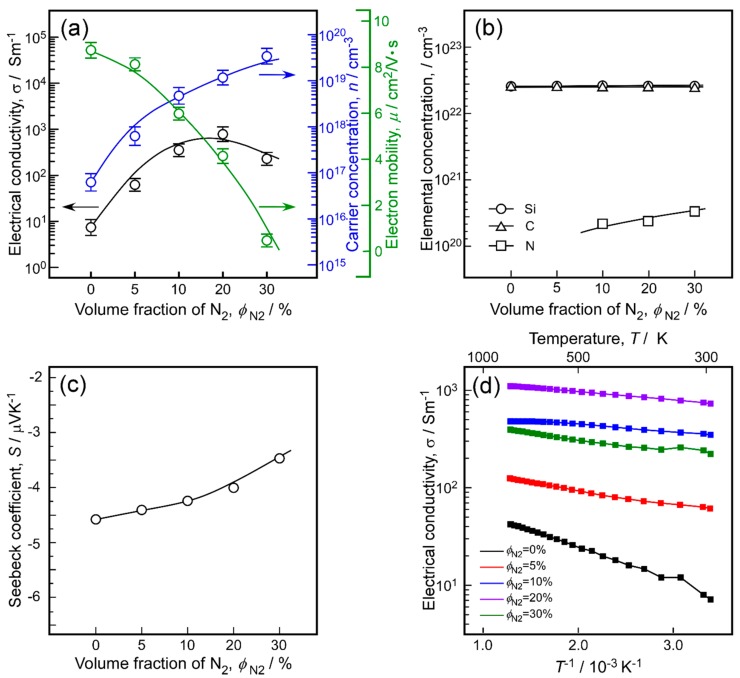
Effect of *ϕ*_N2_ on electrical conductivity, carrier concentration, and mobility (**a**), elemental compositions (**b**), and Seebeck coefficient (**c**) of N-doped 3C-SiC bulks. (**d**) Temperature dependence of electrical conductivities of N-doped 3C-SiC bulks.

**Figure 8 materials-13-00410-f008:**
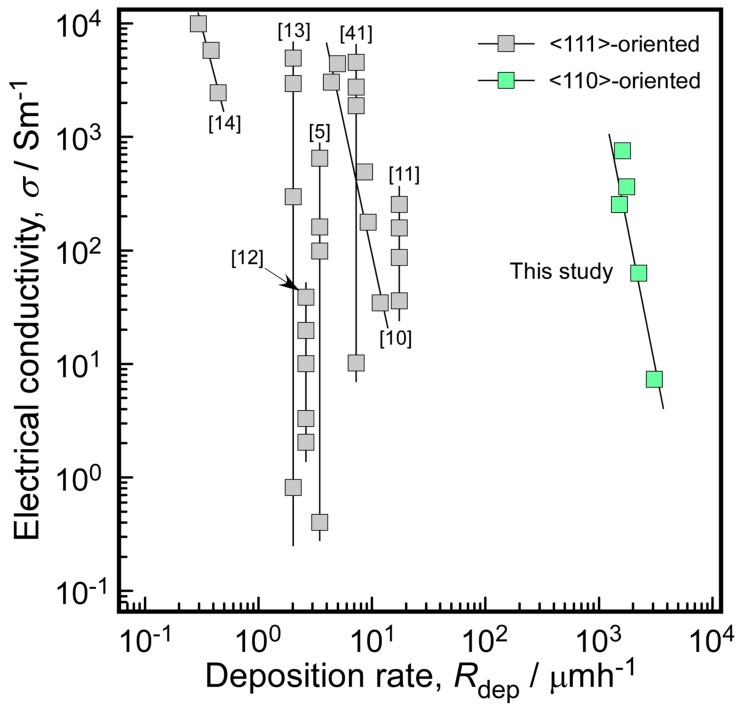
Comparison of electrical conductivities and deposition rates of N-doped 3C-SiC samples prepared via CVD.

**Table 1 materials-13-00410-t001:** Deposition parameters.

Precursor	SiCl_4_ + CH_4_
Diluting gas	H_2_
Dopant	N_2_
Substrate	Graphite
*T* _dep_	1623 K
*P* _tot_	4 kPa
Flow rate of SiCl_4_/CH_4_	600/200 sccm
Flow rate of H_2_	1200 sccm
*ϕ* _N2_	0–30%
Distance between the nozzle and the substrate	30 mm
Deposition time	20 min

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
