# Peer review of "In Situ Doping of Nitrogen in <110>-Oriented Bulk 3C-SiC by Halide Laser Chemical Vapour Deposition"

_materials, 2020, doi:10.3390/ma13020410_

Round 1
Reviewer 1 Report
This work is about “In situ doping of nitrogen in <110>-oriented bulk 3C-SiC by halide laser chemical vapour deposition”. The results of this paper are interesting, which would attract attentions from a wide range of readers. However, there are still some ambiguous. Thus I recommend it to be accepted after some minor revisions. Some suggestion and minor comments are listed below
1. Novelty of the work and the rationale of arriving at the problem is not mentioned clearly. More precise and to the point abstract and conclusions
2. Introduction can be elaborated with significance and more literature review for different aspects. More references are to be added corresponding to composition and individuality of the materials.
3. The author claims that “morphology gradually changed from cauliflower to fourfold-symmetric pyramid”after N2 But it seems a mixture of various shapes, in which a few pyramid shaped structure could be seen in SEM images of Fig. 3 e
4. Spelling and grammatical mistakes and other errors in the manuscript should be carefully checked.
Author Response
Response to Reviewer 1 Comments
Point 1: Novelty of the work and the rationale of arriving at the problem is not mentioned clearly. More precise and to the point abstract and conclusions.
Response 1: Thank you for your kind comment. The novelty and the rationale of arriving at the problem in this paper are mainly the following two points. The first novelty is to obtain N-doped 3C-SiC bulk at high deposition rate by halide laser CVD, since the previous N-doped 3C-SiC are almost in the form of thin films, to further expand its application field. The second novelty is the preparation of <110>-oriented nitrogen-doped 3C-SiC since the doped 3C-SiC films obtained in literature showed an almost exclusive <111> orientation. The Young's modulus of <110>-oriented 3C-SiC (about 350 GPa) are lower than that of <111>-oriented 3C-SiC (about 500 GPa), so it may be used as a better sensor material in MEMS. The rationale was described in detail in the introduction, but it is not well reflected in the abstract. Therefore, we added the information to the last part in abstract. In the conclusion, we describe the variation of orientation with the volume fraction of nitrogen in detail and the comparison of deposition rate with other work. Therefore, we think this conclusion is appropriate for this paper.
Revision: P.1L.15, Abstract: Doping of nitrogen is a promising approach to improve the electrical conductivity of 3C-SiC and allow its application in various fields. N-doped, <110>-oriented 3C-SiC bulks with different doping concentrations were prepared via halide laser chemical vapour deposition (HLCVD) using tetrachlorosilane (SiCl4) and methane (CH4) as precursors, along with nitrogen (N2) as a dopant. We investigated the effect of the volume fraction of nitrogen (fN2) on the preferred orientation, microstructure, electrical conductivity (s), deposition rate (Rdep), and optical transmittance. The preference of 3C-SiC for the <110> orientation increased with increasing fN2. The s value of the N-doped 3C-SiC bulk substrates first increased and then decreased with increasing fN2, reaching a maximum value of 7.4´102 S/m at fN2 = 20%. Rdep showed its highest value (3000 mm/h) for the undoped sample and decreased with increasing fN2, reaching 1437 mm/h at fN2 = 30%. The transmittance of the N-doped 3C-SiC bulks decreased with fN2 and showed a declining trend at wavelengths longer than 1000 nm. Compared with the previously prepared <111>-oriented N-doped 3C-SiC, the high-speed preparation of <110>-oriented N-doped 3C-SiC bulks further broadens its application field.
Point 2: Introduction can be elaborated with significance and more literature review for different aspects. More references are to be added corresponding to composition and individuality of the materials.
Response 2: Thank you for your suggestion. The introduction has been further polished. We removed the sintered SiC parts that are not very relevant to this work, and added a comparison of other component semiconductors prepared by in-situ doping during CVD. In addition, we have added a comparison of the properties of the two preferred orientations of 3C-SiC. Finally, significance of this work to other semiconductor systems has been discussed at the end of introduction.
Revision: P.2L.34, Introduction: Silicon carbide (SiC) is a promising electronic material, which has been applied in various fields due to its excellent mechanical properties, chemical stability, and high durability [1,2]. Although SiC is an intrinsic semiconductor, a high electrical conductivity (s) is a necessary requirement for certain applications, such as solar cells [3], electromagnetic shielding materials [4], pressure sensors [5], and electro-discharge machining [6]. In situ doping via chemical vapour deposition (CVD) is a common method to improve the electrical conductivity of intrinsic semiconductors and allow its application in various fields, such as diamond, ZnO, TiO2, etc [7–9]. In recent decades, a great deal of work has been focused on the preparation of N-doped SiC by adding NH3 [5,10] or N2 [11,12] during CVD, achieving s values from 0.1 to 103 S/m. However, the deposition rates (Rdep) were limited to values between 0.1 and 10 mm/h [10–14]. Due to the low deposition rate, conventional CVD is usually effective for preparing SiC films but not bulk crystals, which greatly limits the applications of doped SiC.
Our research group has recently developed an efficient halide laser CVD (HLCVD) process to fabricate dense and pure 3C-SiC bulks with high Rdep, including transparent 3C-SiC bulks with low defect density and highly preferred orientations [15–17]. The maximum Rdep of 3C-SiC prepared by HLCVD reached 3600 μm/h, which is 10 to 103 times greater than that of conventional CVD. Preferred orientation has a great influence on the properties of corresponding crystals, for example, <111>-oriented 3C-SiC has higher hardness than that of <110>- and <111> co-oriented [15], <110>- 3C-SiC can be used in biosensors by modifying it using an organic molecule [18]. Although 3C-SiC prepared by CVD usually exhibit <111> and <110> preferred orientations, the doped 3C-SiC obtained in previous studies showed an almost exclusive <111> orientation. Nevertheless, the <110>-oriented 3C-SiC may be a more sensitive structural material and could be applied to pressure sensors in microelectromechanical systems (MEMS) [19], due to its lower Young’s modulus(about 350 GPa) than that of the <111>-oriented 3C-SiC(about 500 GPa).
NH3 is an effective and widely used precursor in N-doped SiC, but in the chloride precursor system, especially when the flow rate of the precursor is large, NH3 reacts violently with chloride to form ultra-fine powdery by-products and easily block the pipeline and vacuum pump. In addition, large amounts of Si-N bonds are easily generated due to the high reactivity of NH3. On the contrary, although the reaction activity of nitrogen is low, N2 is often used as the precursor of semiconductor doping because of its good safety and large flow regulation range. In our previous study, we showed that laser CVD can rapidly produce highly <110>-oriented 3C-SiC at a relatively low temperature (approximately 1523 K) [15]. To obtain a <110>-oriented 3C-SiC bulk with high electrical conductivity at high Rdep, in situ nitrogen doping was conducted by adding nitrogen during the preparation of 3C-SiC via HLCVD. In this study, we prepared thick, conductive, and N-doped <110>-oriented 3C-SiC bulks and investigated the effect of the volume fraction of N2 (fN2) on their preferred orientation, deposition rate, electrical conductivity, and optical transmittance. This study provides a new promising route for the rapid fabrication of various doped bulk semiconductors with specific preferred orientation.
Point 3: The author claims that “morphology gradually changed from cauliflower to fourfold-symmetric pyramid”after N2 But it seems a mixture of various shapes, in which a few pyramid shaped structure could be seen in SEM images of Fig. 3 e.
Response 3: Thank you for your comment. In our previous study, it is shown that the pyramid structure often exists in completely <110>-oriented 3C-SiC. Figure 3 shows that the morphology of 3C-SiC changes from cauliflower to fourfold-symmetric pyramid with the increase of nitrogen volume fraction, which means the enhancement of <110>-orientation. However, it is not a complete pyramid structure in Fig. 3(e), indicating that 3C-SiC is still not completely <110>-oriented when nitrogen volume fraction reaches 30%, which is consistent with the XRD results.
Revision: P.7L.153. …… and the morphology gradually changed from cauliflower to fourfold-symmetric pyramid microstructure. The pyramidal grains were considered to be the final stage of the twin-plane propagation model previously proposed for fully <110>-oriented 3C-SiC. The change of morphology indicates the enhancement of <110>-orientation. However, it is not a complete pyramid structure in Fig. 3.e, indicating that 3C-SiC is still not completely <110>-oriented when nitrogen volume fraction reaches 30%, which is consistent with the XRD results. ……
Point 4: Spelling and grammatical mistakes and other errors in the manuscript should be carefully checked.
Response 4: Thank you for your kind comment. The English have been polished by a native English speaker, who belongs to Elsevier Language Editing (Reference: LE258415, Order nr: 231098)
Thank you very much for spending your valuable time. We hope the revision is suitable to be published in Materials.
Sincerely yours,
Rong Tu
Wuhan University of Technology

Reviewer 2 Report
The article is well written and in general, organized and the experiments are presented with high standards. However, the reviewer would like to point out that figure 6 needs to be revised and the focus needs to be adjusted. Please add the photographs and indicate in the caption where each curve a)-e) corresponds to.
Also, if indicated that their potential use is for MEMS (submitted for special issue) due to its lower Young modulus, please measure/simulate and give its (estimated) value. Otherwise, discuss the results presented more towards their application to MEMS.
Author Response
Response to Reviewer 2 Comments
Point 1: The article is well written and in general, organized and the experiments are presented with high standards. However, the reviewer would like to point out that figure 6 needs to be revised and the focus needs to be adjusted. Please add the photographs and indicate in the caption where each curve a)-e) corresponds to.
Response 1: Thank you for your kind suggestions. To avoid unnecessary misunderstandings, the legends was placed on the left side of the photographs, and each curve were marked with the corresponding legends with an arrow, as shown in the following figure (The figure on the left is the original figure, and the right is the modified one). The caption of figure 6 have been changed accordingly.
Fig. 6 Photograph and optical transmission curves of mirror-polished N-doped 3C-SiC bulks with fN2 = 0% (a), 5% (b), 10% (c), 20% (d), and 30% (e).
Point 2: Also, if indicated that their potential use is for MEMS (submitted for special issue) due to its lower Young modulus, please measure/simulate and give its (estimated) value. Otherwise, discuss the results presented more towards their application to MEMS.
Response 2: Thank you for this comment. Ref. 16 reported that the Young's modulus of highly <110>-oriented 3C-SiC values from 340 to 357 GPa, but poly-SiC films having a mixture of both <110>- and <111>- orientations had Young’s modulus values ranging from 452 to 494 GPa. Therefore, the Young's modulus of <110>-oriented 3C-SiC is at least 25% lower than that of <111>-oriented 3C-SiC.
Revision: P.3L.59, Nevertheless, the <110>-oriented 3C-SiC may be a more sensitive structural material and could be applied to pressure sensors in microelectromechanical systems (MEMS) [16], due to its lower Young’s modulus(about 350 GPa) than that of the <111>-oriented 3C-SiC(about 500 GPa).
Thank you very much for spending your valuable time. We hope the revision is suitable to be published in Materials.
Sincerely yours,
Rong Tu
Wuhan University of Technology

Reviewer 3 Report
The manuscript presents work on N doping of SiC to improve conductivity. N2 was used as a precursor. In general manuscript is well organized and is scientifically sound. I have some minor edit requests that I believe would make the work more thorough. If the authors address these, I strongly recommend publishing this manuscript.
Please elaborate regarding why N2 was used as a precursor and contrast with other works.
Please comment regarding what polytypes are forming.
Could any of the results be explained by formation of different polytypes as opposed to N doping?
In figure 6 please indicate which plot corresponds to which volume fraction of N2.
Author Response
Response to Reviewer 3 Comments
Point 1: Please elaborate regarding why N2 was used as a precursor and contrast with other works.
Response 1: Thank you for your kind suggestions. The reason why N2 was used as a precursor instead of NH3 was added to the third paragraph of the introduction.
Revision: P.3L.64, NH3 is an effective and widely used precursor in N-doped SiC, but in the chloride precursor system, especially when the flow rate of the precursor is large, NH3 reacts violently with chloride to form ultra-fine powdery by-products and easily block the pipeline and vacuum pump. In addition, large amounts of Si-N bonds are easily generated due to the high reactivity of NH3. On the contrary, although the reaction activity of nitrogen is low, N2 is often used as the precursor of semiconductor doping because of its good safety and large flow regulation range. In our previous study, we showed that laser CVD can rapidly produce highly <110>-oriented 3C-SiC at a relatively low temperature (approximately 1523 K) [13]. To obtain a <110>-oriented 3C-SiC bulk with high electrical conductivity at high Rdep, in situ nitrogen doping was conducted by adding nitrogen during the preparation of 3C-SiC via HLCVD……
Point 2: Please comment regarding what polytypes are forming.
Response 2: Thank you for your comment. In this study, XRD results showed that only one polytypes of SiC appeared, i.e., 3C-SiC. The other polytypes, e.g., hexagonal SiC, have not detected.
Point 3: Could any of the results be explained by formation of different polytypes as opposed to N doping?
Response 3: Thank you for your kind suggestions. In our study, the polytype of SiC was not affected by the flow rate of nitrogen but the orientation of SiC was affected (<110>-orientation increased). We try to give the reasons for this phenomenon in the first paragraph of the results and discussion section (P.6L.133), as follows:
Revision: P.6L.133, ……To the best of our knowledge, this is the first report of N-doped 3C-SiC with <110> orientation. According to Refs. [21,22], <111>-oriented 3C-SiC shows a polar crystal structure containing Si {111} and C {-1-1-1} planes. The substituted N atoms mostly occupy the C sites in the SiC lattice. However, the adsorption of C or N on Si atoms along the Si-terminated surface of <111>-oriented 3C-SiC grains during the growth is relatively weak. With increasing fN2, the site competition between C and N atoms becomes more intense, which may reduce the growth rate of the <111> orientation and result in the increase of F110. ……
Point 4: In figure 6 please indicate which plot corresponds to which volume fraction of N2.
Response 4: Thank you for your kind comment. Figure 6 and corresponding figure caption have been changed accordingly.
Fig. 6 Photograph and optical transmission curves of mirror-polished N-doped 3C-SiC bulks with fN2 = 0% (a), 5% (b), 10% (c), 20% (d), and 30% (e).
Thank you very much for spending your valuable time. We hope the revision is suitable to be published in Materials.
Sincerely yours,
Rong Tu
Wuhan University of Technology
